# Five-Year Analysis of Microbial Keratitis Incidence, Isolates, and In Vitro Antimicrobial Sensitivity in the South West of England: An Epidemiological Study

**DOI:** 10.3390/microorganisms13071578

**Published:** 2025-07-04

**Authors:** Poonam Sharma, Chimwemwe Chipeta, Kieran O’Kane, Alexander Whiteman, Bryher Francis, Richard Thornton, Indy Sian, Charlotte Buscombe, Jennifer Court, Nathaniel Knox-Cartwright, Harry Roberts

**Affiliations:** 1West of England Eye Unit, Royal Devon University Healthcare NHS Foundation Trust, Exeter EX2 5DW, Devon, UK; poonam.sharma7@nhs.net (P.S.); charlotte.buscombe3@nhs.net (C.B.); n.knoxcartwright@nhs.net (N.K.-C.); 2Vision and Eye Research Institute, Anglia Ruskin University, Cambridge CB1 2LZ, Cambridgeshire, UK; c.chipeta@nhs.net; 3Derriford Hospital, University Hospitals Plymouth NHS Foundation Trust, Plymouth PL6 8DH, Devon, UK; kieran.okane@nhs.net; 4Royal Cornwall Hospital, Royal Cornwall Hospitals NHS Trust, Truro TR1 3LJ, Cornwall, UK; alexander.whiteman@nhs.net; 5Torbay Hospital, Torbay and South Devon NHS Foundation Trust, Torquay TQ2 7AA, Devon, UK; bryher.francis@uhbw.nhs.uk (B.F.); j.court1@nhs.net (J.C.); 6North Devon District Hospital, Royal Devon University Healthcare NHS Foundation Trust, Barnstaple EX31 4JB, Devon, UK; richard.thornton3@nhs.net; 7Musgrove Park Hospital, Somerset NHS Foundation Trust, Taunton TA1 5DA, Somerset, UK; indy.sian@somersetft.nhs.uk; 8Faculty of Health and Life Sciences, University of Exeter, Exeter EX1 2HZ, Devon, UK

**Keywords:** microbial keratitis, culture, empirical, antibiotic, sensitivity

## Abstract

To determine the incidence, causative organisms, and treatment effectiveness for microbial keratitis (MK) in the Southwest of England. Retrospective analysis of 872 corneal scrapes (January 2018–December 2022). Microbiology results were evaluated for organism growth and antimicrobial sensitivity. Data were divided into two groups for trend analysis (A: 2018–2020, B: 2021–2022). Of the 872 scrapes, 357 (39.6%) were culture positive. Bacteria accounted for 90.2% of cases, followed by viruses (2.8%), fungi (2.5%), mixed bacterial growth (2.5%), and Acanthamoeba (2.0%). The estimated incidence of MK was 9.69/100,000/year. Group B had a significantly higher overall MK incidence, with no change in pathogen distribution. *Pseudomonas aeruginosa* was the most frequent isolate (69 cases, 19.3%). In vitro sensitivity to fluoroquinolones was 94.4% for Gram-positive and 98.6% for Gram-negative bacteria. All fungal isolates were sensitive to at least one antifungal. Bacterial pathogens dominate MK in the Southwest of England, with over 90% sensitivity to chloramphenicol, fluoroquinolones, and aminoglycosides, indicating low antimicrobial resistance. Fluoroquinolones remain the recommended first-line therapy for MK. Fungal and protozoal keratitis are rare (<3% of cases), supporting bacteria-focused empirical treatment with close monitoring.

## 1. Introduction

Microbial keratitis (MK) represents a significant cause of ocular morbidity globally [1,2,3]. In the UK alone, up to 3.3% of eye casualty attendances are for MK, and it is responsible for over 5% of blindness globally [1,2,3,4]. The prevalence of causative organisms varies between countries and between regions within the same country [3,4,5]. The true burden of microbial keratitis remains unknown due to underreporting in deprived areas and inaccurate data reporting [1].

Established risk factors for MK include contact lens wear, poor ocular surface (keratitis sicca, systemic disease, neurotrophic cornea secondary to previous herpetic infection, diabetes, or exposure keratopathy), previous ocular surgery/trauma, and exposure to intraocular pressure-lowering drops [6,7,8].

Initial treatment regimens in the UK commonly feature broad-spectrum antibacterial therapy, aimed at covering the diverse range of bacterial organisms associated with MK [4,9]. Many trusts opt for fluoroquinolone monotherapy due to its excellent broad-spectrum coverage, high ocular penetration and tear film concentration, ease of access and storage, and low ocular surface toxicity [4,9,10]. This empirical therapy should ordinarily target the most prevalent causative bacterial microorganisms in the local region. Therapy can later be adapted based on microscopy, culture, and sensitivity (MCS) reports, although it may take up to 2 weeks to yield results in some cases. Sensitivity is defined as inhibition of growth of an organism on exposure to an antimicrobial agent.

Antimicrobial resistance and the emergence of difficult-to-treat species threaten to render existing empirical treatments ineffective [9,10,11]. This issue highlights the importance of auditing corneal scrape data.

There have been multiple large-scale retrospective studies of corneal scrapes across the UK, but not within the Southwest of England (limited to Cornwall, Devon, and Somerset). In this study we present the results of 5-year data collection from corneal scrapes taken in the Southwest of England, with the hope that it will be of benefit to ophthalmic clinicians, nurse practitioners, and prescribing optometrists in the UK.

### 1.1. Aetiology and Epidemiology of Microbial Keratitis

Bacterial organisms are the most common cause of microbial keratitis in developed countries, whereas fungal organisms prevail in the developing world [1,12,13]. Viral keratitis is typically not considered a form of microbial keratitis, despite being a type of infectious keratitis, as viruses are not considered living organisms [1,13].

The most common organisms reported in the literature include *Pseudomonas aeruginosa* and *Staphylococcus* species [3,12,13,14].

A retrospective study published in Nottingham in 2024 found that *Pseudomonas aeruginosa* and *Staphylococcus* species were the most common pathogens isolated in patients with contact lens-related bacterial keratitis (CLBK) and were the causative organisms for approximately 50% and 20% of all CLBK with a positive scrape result, respectively [14]. The Nottingham Infectious Keratitis Study (2021) [4] showed that *Pseudomonas aeruginosa* was the single most commonly isolated organism in all infectious keratitis (not solely contact lens-related), despite Gram-positive bacteria accounting for 53.8% of organisms isolated.

The East of England microbial keratitis study, which examined the incidence of microbial keratitis from 2015 to 2020, found that *Pseudomonas* spp. and *Staphylococcus aureus* were the two most common pathogen groups identified from corneal scrapes and were present in 29.57% and 13.04% of all positive scrapes [3].

Older studies, including the Portsmouth corneal ulcer study (2009) [6] and the Oxford study (2011) [9], had similar findings. *Staphylococcus epidermidis*, *Pseudomonas aeruginosa*, and *Staphylococcus aureus* were the most prevalent species isolated from corneal scrapes in Portsmouth [6]. These organisms were found on 31.7%, 12%, and 11.5% of all positive scrapes, respectively. Interestingly, they found that Gram-positive bacteria accounted for the majority of contact lens-related keratitis (CLRK), in contrast to other studies. In the Oxford study, *coagulase-negative Staphylococci*, *Pseudomonas aeruginosa*, and *Staphylococcus aureus* were the most frequently encountered species, comprising 24.8%, 24.3%, and 12.4% of the cultures, respectively [9].

Obtaining accurate data on the true epidemiology of microbial keratitis remains a challenging task for numerous reasons [1,12,13]. Infection rates are often underreported, especially in developing countries where access to healthcare can be limited. Even in developed countries, rates are likely to be underestimated due to low detection rates of microscopy, culture, and stain (MCS) [1]. Furthermore, data is often reported under the umbrella term “corneal blindness”, thereby making it impossible to discern the true cause of vision loss [1,12].

### 1.2. Antimicrobial Susceptibility

The use of broad-spectrum antibacterial agents is supported by numerous studies in the literature.

Suresh et al. (Nottingham, 2024) showed that Gram-positive bacteria demonstrated the highest susceptibility to vancomycin (100%) and aminoglycosides (>94%), and Gram-negative bacteria demonstrated the highest susceptibility to aminoglycosides (>95%). Susceptibility to fluoroquinolones was lower, at 50–77.8% for Gram-positive bacteria and 75% for Gram-negative bacteria [14].

Reassuringly, the East of England microbial keratitis study (2023) [3] demonstrated excellent response to fluoroquinolones, with common bacteria such as *Streptococcus* species and *Staphylococcus aureus* being 100% susceptible, despite other Gram-positive bacteria being less so [3]. Gram-negative bacteria showed excellent susceptibility to fluoroquinolones, at 97.2%. Gram-positive bacteria overall were found to be more susceptible to aminoglycosides and glycopeptides.

Fluoroquinolones were shown to have excellent coverage against both Gram-positive and Gram-negative bacteria in the Nottingham infectious keratitis study and the Oxford study [4,9]. In both studies, fluoroquinolones were more effective against Gram-negative bacteria. In the Nottingham study, in vitro antimicrobial susceptibilities for cephalosporin, fluoroquinolone, and aminoglycoside were 100.0%, 91.9%, and 95.2% for Gram-positive bacteria and 81.3%, 98.1%, and 98.3% for Gram-negative bacteria. In the Oxford study, in vitro antimicrobial susceptibilities for cephalosporin, fluoroquinolone, and aminoglycoside were 80.9%, 85.4%, and 87.2% for Gram-positive bacteria and 48.1%, 99.0%, and 100.0% for Gram-negative bacteria.

The Nottingham infectious keratitis study concluded that both first-line treatment options of combined cephalosporin/aminoglycoside and fluoroquinolone monotherapy were suitable to use in patients presenting with MK. The four most commonly isolated bacteria demonstrated over 90% susceptibility to either regimen. The Oxford study concluded that the combination of gentamicin and cefuroxime was significantly more likely to be active than ciprofloxacin alone but that fluoroquinolone monotherapy was a reasonable first-line treatment due to its good broad-spectrum coverage.

## 2. Material and Methods

### 2.1. Data Collection

This is a retrospective epidemiological study of all patients diagnosed with microbial keratitis who subsequently underwent corneal scraping within a 5-year period. Socioeconomic data was not included due to lack of availability from the documentation. The medical records of patients who presented with microbial keratitis to the eye casualty departments of participating centres between 1 January 2018 and 31 December 2022 were reviewed. All those who underwent corneal scraping were included. Patients who were empirically treated without corneal scraping were excluded. Participating hospitals included Royal Devon and Exeter Hospital, Torbay Hospital, Musgrove Park Hospital, North Devon District Hospital, and Royal Cornwall Hospital.

Population size (of areas covered by each hospital) from 2018 to 2022 was estimated using data from the Office of National Statistics (ONS). This was used to calculate the incidence of MK.

### 2.2. Hospital Scraping Protocols

All patients who presented with a corneal ulcer measuring ≥ 1 mm in diameter in any part of the cornea or an ulcer measuring ≥ 0.5 mm within the pupillary zone (central 3 mm) of the cornea underwent corneal scraping. Samples were taken from the base or edge of the ulcer using either a sterile needle or scalpel. A new instrument was used for each agar plate, slide, or vial. Scrapes were inoculated onto a microscopy slide for Gram stain and 4 plates: blood agar, chocolate agar, fastidious anaerobe agar, and Sabouraud dextrose agar (fungi). For patients in whom viral or Acanthamoeba keratitis was suspected, a sample was sent for polymerase chain reaction (PCR) analysis using the approved collection method at each hospital.

Data were collected pro forma, including hospital trust, number, date of scrape, Gram stain, isolates, and antimicrobial sensitivities. Patient age and gender were recorded, but no other demographic data was collected. Organisms were classified as Gram-positive bacteria, Gram-negative bacteria, Acanthamoeba, viruses, or fungi. Gram-positive and Gram-negative bacteria were identifiable from the culture slides sent as a part of the corneal scrape test. Fungal organisms were identified from samples sent on the Sabouraud’s agar plate. Polymicrobial keratitis was defined as MK caused by ≥2 organisms from a single episode of infection.

At Torbay Hospital, scrapes were also inoculated in brain–heart infusion broth, and presumed bacterial ulcers were swabbed for HSV to exclude concomitant viral infection. Corneal scraping technique was strictly monitored at the North Devon District Hospital; samples were taken in the presence of a microbiologist to ensure adequate scraping protocol was followed.

No additional broths or plates other than those mentioned above were used at the other hospitals included in this study. It is not routine practice to swab all ulcers for HSV at the Royal Devon and Exeter Hospital, Musgrove Park Hospital, North Devon District Hospital, or Royal Cornwall Hospital.

### 2.3. Hospital Treatment Protocols

All five eye departments included in this study prescribe hourly levofloxacin monotherapy as first-line treatment for microbial keratitis.

### 2.4. Ethical Approval

Ethical approval was not required for this retrospective study. This study was conducted in accordance with the tenets of the Declaration of Helsinki.

### 2.5. Statistical Analysis

For ease of trend analysis, and in keeping with other similar studies [3,4,9], the data were split into two groups: 2018–2020 (Group A; pre-COVID-19) and 2021–2022 (Group B, post-COVID-19). The time period covered included the 3 years preceding and the 2 worst years of the COVID-19 global pandemic; therefore, we wished to determine if the lockdown led to a decrease in the incidence of MK cases.

For statistical analysis the SciPy and Statsmodels libraries were used in the Python 3.11 programming language (Python™).

The data were not continuous and therefore did not need to be assessed for normality.

## 3. Results

### 3.1. Corneal Scrape Outcomes and Incidence of Microbial Keratitis

A total of 872 corneal scrapes were performed in patients with MK with a mean age of 56.8 (range 4.0–97.0). A total of 52.8% (460) were female and 47.2% (412) were male. The mean incidence of MK over the 5 years was estimated at 9.69 per 100,000 per year (range 7.37–13.40, Figure 1). Incidence was based on scrape numbers only. There was no consistent upward trend on regression analysis of incidence over the 5-year period. On group comparison, however, there was a significantly higher incidence of MK in group B (*p* < 0.0001).

### 3.2. Analysis of Organisms Isolated

Of the 873 scrapes, 357 (39.58%) were culture positive. The remainder either did not grow organisms on culture media, or the laboratory results failed to give further clarification on laboratory findings. Gram-positive bacteria were the most commonly isolated organisms, making up 53.8% of positive isolates (Table 1). Gram-negative bacteria accounted for a further 36.4% of positive isolates, followed by viruses (2.8%), mixed bacterial growth and fungi (both 2.5%), and lastly Acanthamoeba (2.0%). There were 198 sensitivities available. The average number of antimicrobials tested per positive culture was 5.68 (range 1–12).

The most frequently isolated pathogens were *Pseudomonas aeruginosa* (69 cases, 19.33%), coagulase-negative *staphylococci* (CoNS) (56 cases, 15.69%), *Staphylococcus aureus* (44 cases, 12.32%), and *Streptococcus* spp. (26 cases, 7.28%) (Figure 2). Nine cases were polymicrobial in origin, of which three grew both Gram-positive and Gram-negative bacteria. No samples were positive for mixed bacteria/fungi or mixed bacteria/Acanthamoeba growth.

The proportion of Gram-positive bacteria isolated ranged from 49.1% to 58.8% of the total number of pathogens isolated per year. For Gram-negative bacteria, this ranged from 27.9% to 40.4% (Table 1). The rate of detection of almost all bacterial organisms did not display an increasing or decreasing trend throughout the 5 years. *Staphylococcus epidermidis* was the sole organism with a significant difference in incidence between the two groups (group A: 2.1%, group B: 6.3%; *p* < 0.0001).

### 3.3. Antimicrobial Sensitivities

Gram-positive. In vitro sensitivities to chloramphenicol, fluoroquinolones, aminoglycosides, and cephalosporins were 96.3% (52/54), 94.4% (51/54), 93.3% (56/60), and 78.8% (26/33), respectively. Average sensitivity of Gram-positive organisms to these antibiotic classes was 92.0% (185/201). There was no significant difference in sensitivity levels or resistance profiles between the two groups (Table 2).

Gram-negative. In vitro sensitivities to chloramphenicol, fluoroquinolones, and aminoglycosides were 96.6% (28/29), 98.6% (68/69), and 100.0% (54/54), respectively. Average sensitivity of Gram-negative organisms to these antibiotic classes was 98.7% (150/152). Once again there was no significant difference in sensitivity or resistance between the two groups (Table 2). There were no available sensitivities to cephalosporins in the Gram-negative group.

On analysis of in vitro sensitivities of individual bacterial species or genii, the Gram-positive organisms *Staphylococcus aureus*, *Staphylococcus epidermidis,* and *Streptococcus* spp. showed 100% sensitivity to chloramphenicol. *Bacillus* spp. showed 87.5% sensitivity to chloramphenicol. These organisms also showed high rates of sensitivity to fluoroquinolones. Gentamicin was less effective against streptococci but showed excellent coverage against *staphylococci*, *streptococci*, and *bacilli*. Fusidic acid showed poor coverage overall, and cephalosporins were extremely effective against *Staphylococcus aureus*, *streptococci*, and *bacilli*. *Staphylococcus epidermidis* was not tested with cephalosporins. *Pseudomonas aeruginosa*, the most prevalent organism isolated, showed excellent susceptibility to fluoroquinolones, cephalosporins, and gentamicin, but not to chloramphenicol and fusidic acid. Overall, fluoroquinolones offer good broad-spectrum coverage. Table 3 shows sensitivities of bacterial groups to commonly used antibacterial agents. There was no significant difference in antimicrobial susceptibility of any pathogen between groups A and B.

Fungal. Of the five fungal sensitivities available, one was resistant to voriconazole, but this same fungal organism was susceptible to natamycin.

## 4. Discussion

Microbial keratitis (MK) remains a significant cause of ocular morbidity throughout the world. This is the first study reporting incidence, isolate profiles, and their antimicrobial sensitivity in the South West of England (not including Bristol and Bath [14]).

### 4.1. Corneal Scrape Outcomes

We found that 39.58% of corneal scrapes were positive. This figure is consistent with data generated from other studies in the UK which show a range of 32.6% to 63.8% [3,4,6,9,15,16,17]. The Portsmouth Study gave the highest positive yield of 63.8% [6].

Despite the scraping technique being monitored at North Devon District Hospital, only 28 of 102 scrapes (27.5%) yielded a positive culture.

Possible explanations for the low detection rates include early presentation with a low number of colony-forming units (CFUs), poor scraping/plating technique, pre-existing treatment with an over-the-counter antibacterial agent prescribed by a GP or pharmacist, or sample collection after starting empirical therapy in an eye unit. In countries such as India, where scrape yield is much higher, hospitals with eye departments are difficult to access from rural areas, which may be thousands of kilometres away from an eye centre [1,2]. Patients may present later in the disease process, with larger ulcers and increased CFUs. Reduced access to healthcare based on location exists in the UK too, but the UK is a comparatively small country.

A lack of pathologists specialising in ocular disease may also be a contributing factor to the low detection rates. Pathology specialist training in the UK lasts 5 years, after a 2-year core medical training programme. The 5-year specialist pathway does not include a module in ocular pathology, according to the Royal College of Pathologists website [18]. Given that up to 3.3% of eye casualty attendances in the UK are for MK [3,4] and that some departments send corneal scrapes from up to 70.2% of MK patients [6], it follows that specialist training in ocular pathology should be offered as a part of the pathology-training pathway or that pathologists who routinely examine ocular specimens should receive training in this discipline.

### 4.2. Incidence of Microbial Keratitis

The incidence of MK decreased in the two years leading up to the start of the COVID-19 pandemic and the first UK lockdown. Unsurprisingly, the incidence of MK was at its lowest in 2020, much of which was spent in lockdown. Possible explanations include fewer people wearing contact lenses due to (a) an inability to obtain them from optometrists or (b) choosing not to wear them due to lack of socialisation. Incidence rose again after lockdown, more sharply in 2022, possibly reflecting greater contact lens wear in general and in unsanitary locations, such as showers and swimming pools. The incidence of MK in 2022 was higher than pre-COVID-19 levels. This may reflect an increasing trend in contact lens wear, especially amongst younger people, in addition to a rise in correlation with resumption of post-COVID-19 social activities.

Contact lens wear has been shown to increase the risk of MK (especially of bacterial aetiology) [6,19]. Bacteria may become sequestered between the posterior surface of the contact lens and the corneal epithelium, which are then allowed to stagnate in the absence of tear film washing over the corneal epithelium. The risk of MK can be up to 80 times higher in contact lens wearers compared to non-contact lens wearers [6,20].

Incidence of MK varies widely across the UK, with incidence being as low as 3.6 per 100,000/year in the West of Scotland and as high as 34.7 per 100,000/year in Nottingham [4,20]. Our study found a rate on the lower end of the spectrum at 9.69 per 100,000/year. This is likely to be an underestimation of the true incidence for several reasons. Firstly, not all corneal ulcers are scraped in a clinic; often ulcers with an epithelial defect measuring less than 1.0 mm in diameter or those in the periphery are simply treated empirically. Given that the majority improve with empirical treatment, there is no need to scrape. Secondly, over-the-counter treatments such as chloramphenicol or fusidic acid are often given first-line in the community for any red eye presentation. As shown by the data, there is high sensitivity to both agents amongst both Gram-positive and Gram-negative organisms. Chloramphenicol alone was shown to be 93.6% and 96.6% effective against Gram-positive and Gram-negative organisms, respectively. Patients who were treated successfully with first-line agents would not have necessarily presented to an eye clinic or would not have been scraped had they presented with a resolving ulcer.

### 4.3. Isolate Analysis

The prevalence of distinct pathogenic microbes varies across the globe, affected by factors such as climate and population demographics. This is seen with microbial keratitis, as reported by multiple studies [1,2,5]. In parts of South Asia, MK has reached endemic levels, with incidence being as high as 799 per 100,000 in Nepal [1]. The highest proportions of bacterial aetiology have been reported in Europe, North America, Australia, and Oceania, where contact lens wear is more common [1]. In developing countries, fungal organisms are responsible for an equal proportion of MK cases and even exceed bacterial cases in some regions [1,21].

In this study 90.2% (322) of cultures were positive for bacteria (Gram-positive 53.8% (192), Gram-negative 36.4% (130)), in keeping with data published in Ung and colleagues’ review article [1]. Fungal organisms accounted for 2.5% of positive scrapes. Other UK studies have also reported a similar proportion of both bacterial and fungal keratitis [3,4,6,9,15,16].

*P. aeruginosa* was the most isolated organism overall, comprising 46.8% of Gram-negative organisms and 19.33% of all organisms. *P. aeruginosa* is currently the leading cause of contact lens-related microbial keratitis in the UK, and contact lens wear is the biggest risk factor for microbial keratitis in the UK, so naturally it follows that *P. aeruginosa* should be the most prevalent causative organism [21].

Coagulase-negative *staphylococci* (CoNS) were the next most prevalent organism group isolated. CoNS are the most frequent blood culture isolates and are predominantly contaminants rather than the true source of infection [22]. It would be reasonable to assume that 15.69% of cultures in this dataset were contaminated and that the causative agent remains unknown. Reasons for contamination of the sample include poor sterile technique during plating, not correctly sealing the plates after taking the sample, and plates becoming contaminated in the laboratory. Care must be taken to ensure adequate sterility. When corneal scrape samples are taken in a clinic, there is no sterile field like there is in theatres. Providing a separate trolley in the clinic for corneal scraping, which is cleaned prior to taking a sample, may reduce contamination rates. Needles or scalpels for scraping could be stored in a separate area from the usual clinic equipment. Clinicians should also clean their hands thoroughly and wear gloves when taking corneal scrapes.

There was no significant difference in the incidence of individual organisms between groups A and B, excluding *Staphylococcus epidermidis*, for which the proportion of cases due to this bacterium increased from 2.1% to 10.3% of Gram-positive organisms (*p* = 0.021) (Table 4). *Staphylococcus epidermidis* is a skin commensal which has been shown to cause infections in contact lens wearers [4,6,7,14]. The significant increase in MK due to *Staphylococcus epidermidis* may be related to an increase in contact lens wear following resumption of normal social activities post-lockdown.

### 4.4. Antimicrobial Sensitivities

There were high levels of in vitro sensitivity (>90.0%) to chloramphenicol, fluoroquinolones, and aminoglycosides in both the Gram-positive and negative groups (Table 3). Sensitivity to cephalosporins was relatively low (78.8%) in comparison to the Gram-positive group. There were no significant differences in antimicrobial susceptibility of any bacterial pathogen between groups A and B, thus showing that antimicrobial resistance did not increase between 2018 and 2022 in the South West of England.

Analysis of separate bacterial species (Table 3) showed consistently high sensitivity to fluoroquinolones. Chloramphenicol did not offer poor coverage against *P. aeruginosa*, the most common causative agent of microbial keratitis. Gentamicin offered reasonable coverage against *Streptococcus* species. This is in keeping with other UK studies which also found fluoroquinolones to offer excellent broad-spectrum coverage [3,4,6,9].

Thus, fluoroquinolone monotherapy remains an appropriate first-line empirical therapy for MK in the South West of England. Gentamicin 1.5% (an aminoglycoside) is most likely a better option than cephalosporin for a second antibiotic in cases where the infection is not responding well to fluoroquinolones, although a combination of gentamicin and cephalosporin should provide adequate coverage.

### 4.5. Limitations

This study was based solely on the results of corneal scrapes performed in clinic, excluding patients who were diagnosed and treated in the absence of microbiological assays. Therefore, we cannot state what proportion of patients with MK were scraped. For a more accurate estimate of the incidence of MK, a prospective cohort study in which all cases of presumed MK were recorded could be carried out. Alternatively, searching for all cases of microbial keratitis in hospital records rather than analysing scrape results only might be a better way to estimate incidence.

No socioeconomic data was collected for this study. This data would enable analysis of socioeconomic factors and their impact on MK rates. Similarly, insufficient information on contact lens wear was collected.

There was a low microbe detection rate of the corneal scrape tests performed. Low yield is common in studies evaluating causes of microbial keratitis in the UK. Reasons for low yield have been discussed above.

University Hospitals Plymouth NHS Trust was not included in this study. This exclusion was due to the challenges associated with auditing their microbiology data. Given the hospital’s size and its likely contribution to regional MK cases, its exclusion may have affected the accuracy and completeness of this study’s findings, potentially underestimating the true incidence and diversity of causative organisms in the region.

All cultures that grow bacteria should be inoculated with the same antibiotics to ensure fairness: going forwards, all positive cultures should be tested for sensitivity to fluoroquinolones, cefuroxime, and gentamicin, as these are the readily available antibiotics in hospital pharmacies.

## 5. Conclusions

The estimated incidence of MK in the Southwest of England is 9.69 per 100,000 population/year, although this is most likely an underestimation. Gram-positive organisms are the most widespread cause of MK, although *P. aeruginosa* was the most common organism. First-line agents such as fluoroquinolones provide excellent Gram-positive and Gram-negative coverage, but over-the-counter antimicrobials such as chloramphenicol and fusidic acid remain effective treatment options against common bacterial causes of MK. A combination of gentamicin and cephalosporin is a reasonable second-line option, although Gram-positive organisms display increased resistance to cephalosporins compared to other common antimicrobials. Fluoroquinolone monotherapy should continue to be utilised as first-line treatment in the Southwest of England, similar to other regions of the country.

Further studies looking at contact lens wear and hygiene and demographic and socioeconomic data in the southwest would be useful for determining the most common risk factors for MK in this region.

Hospital laboratories should routinely test scrapes that yield bacteria with ocular antibiotics that are used routinely in clinics in order to effectively guide treatment.

### Key Points

Bacterial organisms account for the majority of microbial keratitis cases in the Southwest of England.

Most (>90.0%) of bacterial organisms are sensitive to fluoroquinolones, which should be continued to be utilised as empirical therapy for MK in the southwest.

## Figures and Tables

**Figure 1 microorganisms-13-01578-f001:**
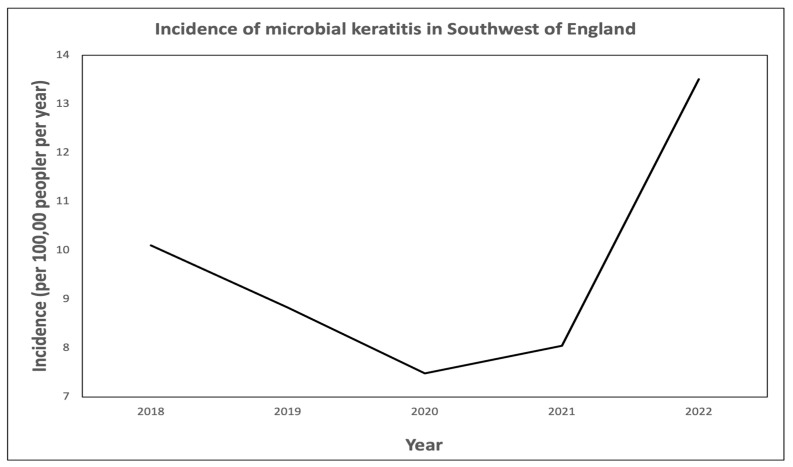
Incidence of microbial keratitis in the Southwest of England.

**Figure 2 microorganisms-13-01578-f002:**
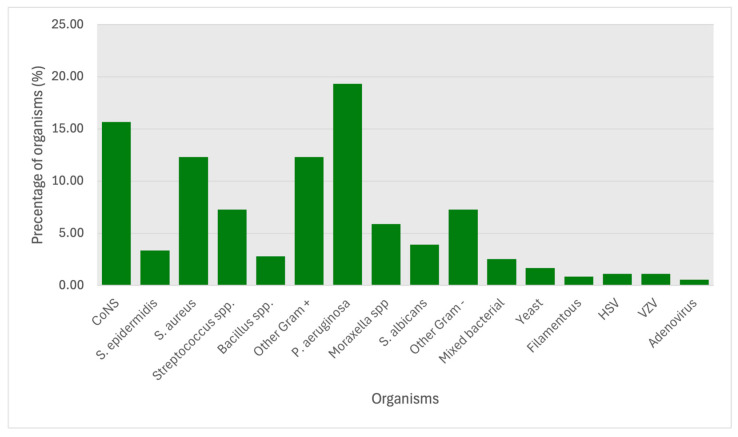
Proportion (%) of microbial isolates cultured in the Southwest of England, 2018–2022.

**Table 1 microorganisms-13-01578-t001:** Incidence of pathogens isolated in the South West of England. Individual pathogens are given as a percentage of the type of organism. CoNS: coagulase-negative *staphylococci*. spp.: species.

	2018		2019		2020		2021		2022		Total	
All Pathogens	68		53		57		76		103		357	
Gram-positive	40	58.8%	27	50.9%	28	49.1%	41	53.9%	56	54.4%	192	53.8%
CoNS	17	42.5%	7	25.9%	7	25.0%	13	31.7%	12	21.4%	56	29.2%
*S. epidermidis*	0	0.0%	1	3.7%	1	3.6%	5	12.2%	5	8.9%	12	6.3%
*S. aureus*	6	15.0%	6	22.2%	10	35.7%	8	19.5%	14	25.0%	44	22.9%
*Streptoccus* spp.	6	15.0%	6	22.2%	3	10.7%	3	7.3%	8	14.3%	26	13.5%
*Bacillus* spp.	2	5.0%	2	7.4%	2	7.1%	2	4.9%	2	3.6%	10	5.2%
Other	9	22.5%	5	18.5%	5	17.9%	10	24.4%	15	26.8%	44	22.9%
Gram-negative	19	27.9%	20	37.7%	23	40.4%	30	39.5%	38	36.9%	130	36.4%
*P. aeruginosa*	9	47.4%	10	50.0%	10	43.5%	17	56.7%	23	60.5%	69	19.3%
*Moraxella* spp.	3	15.8%	5	25.0%	5	21.7%	3	10.0%	5	13.2%	21	5.9%
*S. albicans*	2	10.5%	2	10.0%	3	13.0%	2	6.7%	5	13.2%	14	3.9%
Other	5	26.3%	3	15.0%	5	21.7%	8	26.7%	5	13.2%	26	7.3%
Mixed Growth	1	1.5%	4	7.5%	3	5.3%	1	1.3%	0	0.0%	9	2.5%
Fungi	5	7.4%	2	3.8%	0	0.0%	1	1.3%	1	1.0%	9	2.5%
Yeast	4	80.0%	1	50.0%	0	0.0%	0	0.0%	1	100.0%	6	1.7%
Filamentous	1	20.0%	1	50.0%	0	0.0%	1	100.0%	0	0.0%	3	0.8%
Acanthamoeba	1	1.5%	0	0.0%	1	1.8%	2	2.6%	3	2.9%	7	2.0%
Viruses	2	2.9%	0	0.0%	2	3.5%	1	1.3%	5	4.9%	10	2.8%
*Herpes simplex Virus* (HSV)	2	100.0%	0	0.0%	1	50.0%	1	0.0%	0	0.0%	4	1.1%
*Varicella zoster Virus* (VZV)	0	0.0%	0	0.0%	1	50.0%	0	0.0%	3	60.0%	4	1.1%
*Adenovirus*	0	0.0%	0	0.0%	0	0.0%	0	0.0%	2	40.0%	2	0.6%

**Table 2 microorganisms-13-01578-t002:** Antimicrobial sensitivity of pathogens between 2018–2020 and 2021–2022.

	2018–2020		2021–2022		Total		*p*
	Sensitivity	%	Sensitivity	%	Sensitivity	%	
Gram-positive							
Chloramphenicol	30/32	93.8%	22/22	100.0%	52/54	96.3%	0.818
Ciprofloxacin	29/30	96.7%	22/24	91.7%	51/54	94.4%	0.851
Levofloxacin	7/7	100.0%	4/4	100.0%	11/11	100.0%	1
Ofloxacin	17/18	94.4%	4/4	100.0%	21/22	95.5%	1
All fluoroquinolones	29/30	96.7%	22/24	91.7%	51/54	94.4%	0.851
Fusidic acid	21/29	72.4%	15/22	68.2%	36/51	70.6%	0.859
Gentamicin	32/35	91.4%	23/25	92.0%	55/60	91.7%	0.982
Neomycin	0/0	0.0%	4/4	100.0%	4/4	100.0%	N/A
Tobramycin	4/4	100.0%	0/0	0.0%	4/4	100.0%	N/A
All aminoglycosides	32/35	91.4%	24/25	96.0%	56/60	93.3%	0.857
Vancomycin	5/5	100.0%	18/18	100.0%	23/23	100.0%	1
Teicoplanin	2/2	100.0%	2/2	100.0%	4/4	100.0%	1
Penicillins	31/35	88.6%	33/34	97.1%	64/69	92.8%	0.714
Co-Amoxiclav	1/2	50.0%	2/2	100.0%	3/4	75.0%	1
Tazocin	4/4	100.0%	1/1	100.0%	5/5	100.0%	1
Cephalosporins	20/26	76.9%	6/7	85.7%	26/33	78.8%	0.816
Meropenem	6/6	100.0%	1/1	100.0%	7/7	100.0%	1
Clindamycin	20/25	80.0%	5/7	71.4%	25/32	78.1%	0.821
Macrolides	22/32	68.8%	28/28	100.0%	50/60	83.3%	0.186
Tetracyclines	23/26	88.5%	28/28	100.0%	51/54	94.4%	0.663
Rifampicin	21/21	100.0%	3/4	75.0%	24/25	96.0%	0.16
Linezolid	16/16	100.0%	15/16	93.8%	31/32	96.9%	0.857
Mupirocin	10/14	71.4%	18/18	100.0%	28/32	87.5%	0.391
Co-Trimoxazole	2/2	100.0%	6/8	75.0%	8/10	80.0%	1
Gram-negative							
Chloramphenicol	16/17	94.1%	12/12	100.0%	28/29	96.6%	0.874
Ciprofloxacin	37/38	97.4%	31/31	100.0%	68/69	98.6%	0.913
Levofloxacin	3/3	100.0%	2/2	100.0%	5/5	100.0%	1
Ofloxacin	2/2	100.0%	0/0	0.0%	2/2	100.0%	N/A
Fluoroquinolones	37/38	97.4%	31/31	100.0%	68/69	98.6%	0.913
Fusidic Acid	2/2	100.0%	0/0	0.0%	2/4	50.0%	N/A
Gentamicin	25/25	100.0%	29/29	100.0%	54/54	100.0%	1
Amikacin	4/4	100.0%	7/7	100.0%	11/11	100.0%	1
Tobramycin	5/5	100.0%	2/2	100.0%	7/7	100.0%	1
All aminoglycosides	25/25	100.0%	29/29	100.0%	54/54	100.0%	1
Vancomycin	1/2	50.0%	0/0	0.0%	1/1	100.0%	N/A
Teicoplanin	0/0	0.0%	1/1	100.0%	1/0	100.0%	N/A
Penicillins	6/7	85.7%	4/4	100.0%	10/11	90.9%	1
Co-Amoxiclav	7/9	77.8%	0/0	0.0%	7/9	77.8%	N/A
Tazocin	15/15	107.1%	15/18	83.3%	30/33	90.9%	0.617
Colistin	0/0	0.0%	5/5	100.0%	5/5	100.0%	N/A
Meropenem	20/20	100.0%	22/22	100.0%	42/42	100.0%	1
Clindamycin	1/3	33.3%	12/13	92.3%	13/16	81.3%	0.071
Macrolides	2/3	66.7%	5/5	100.0%	7/8	87.5%	0.375
Tetracyclines	4/5	80.0%	9/9	100.0%	13/14	92.9%	0.357
Rifampicin	2/2	100.0%	1/1	100.0%	3/3	100.0%	1
Daptomycin	0/0	0.0%	4/6	66.7%	4/6	66.7%	N/A
Linezolid	2/2	100.0%	0/0	0.0%	2/2	100.0%	N/A
Mupirocin	2/2	100.0%	0/0	0.0%	2/2	100.0%	N/A
Co-Trimoxazole	0/1	0.0%	0/0	0.0%	0/1	0.0%	N/A
Aztreonam	2/2	100.0%	0/0	0.0%	2/0	100.0%	N/A
Trimethoprim	1/1	100.0%	1/1	100.0%	2/2	100.0%	1
Ticarcillin/Clavulan	1/2	50.0%	2/2	100.0%	3/4	75.0%	1
Fungal							
Voriconazole	0/0	0.0%	1/2	50.0%	1/2	50.0%	N/A
Caspofungin	0/0	0.0%	1/1	100.0%	1/1	100.0%	N/A
Fluconazole	0/0	0.0%	1/1	100.0%	1/1	100.0%	N/A
Amphotericin B	0/0	0.0%	1/1	100.0%	1/1	100.0%	N/A

**Table 3 microorganisms-13-01578-t003:** Sensitivities of different bacterial species to the most commonly prescribed topical antibiotics. CoNS: coagulase-negative *Staphylococcus aureus* (likely contaminant).

Pathogen	Chloramphenicol		Fusidic Acid		Fluoroquinolones		Cephalosporins		Gentamicin	
	Sensitivity	%	Sensitivity	%	Sensitivity	%	Sensitivity	%	Sensitivity	%
Gram-positive	103/109	94.5	57/89	64.0	174/182	95.6	15/15	100.0	81/89	91.0
CoNS	38/42	90.5	29/54	53.7	103/107	96.3	5/5	100.0	49/54	90.7
*S. epidermidis*	2/2	100.0	2/4	50.0	7/7	100.0	N/A		4/4	100.0
*S. aureus*	31/31	100.0	26/29	89.7	53/54	98.1	4/4	100.0	31/33	93.4
*Streptococcus* spp.	18/18	100.0	1/4	25.0	18/20	90.0	2/2	100.0	3/5	60.0
*Bacillus* spp.	7/8	87.5	2/8	25.0	16/16	100.0	2/2	100.0	8/8	100.0
Other	14/15	93.3	9/10	90.0	17/17	100.0	2/2	100.0	8/8	100.0
Gram-negative	41/52	78.8	3/19	15.8	77/79	97.5	35/38	92.1	32/33	96.9
*P. aeruginosa*	1/12	8.3	0/11	0.0	25/25	100.0	10/11	90.0	12/12	100.0
*Moraxella* spp.	17/17	100.0	2/2	100.0	22/22	100.0	6/6	100.0	5/5	100.0
*S. albicans*	7/7	100.0	7/7	100.0	10/10	100.0	5/8	62.5	6/6	100.0
Other	16/16	100.0	1/5	20.0	20/22	90.9	13/13	100.0	9/10	90.0

**Table 4 microorganisms-13-01578-t004:** Microbiological profiles of MK between 2018–2020 and 2021–2022. CoNS: coagulase-negative *Staphylococcus epidermidis*. spp.: species.

	2018–2020		2021–2022		Total		*p*
All Pathogens	178		179		357		
Gram-positive	95/178	53.4%	97/179	54.2%	192/357	53.8%	0.916
CoNS	31/95	32.6%	25/97	25.8%	56/192	29.2%	0.411
*S. epidermidis*	2/95	2.1%	10/97	10.3%	12/192	6.3%	0.021
*S. aureus*	22/95	23.2%	22/97	22.7%	44/192	22.9%	0.985
*Streptococcus* spp.	15/95	15.8%	11/97	11.3%	26/192	13.5%	0.424
*Bacillus* spp.	6/95	6.3%	4/97	4.1%	10/192	5.2%	0.521
Other	19/95	20.0%	25/97	25.8%	44/192	22.9%	0.376
Gram-negative	62/178	34.8%	68/179	38.0%	130/357	36.4%	0.621
*P. aeruginosa*	29/62	46.8%	40/68	58.8%	69/130	53.1%	0.193
*Moraxella* spp.	13/62	21.0%	8/68	11.8%	21/130	16.2%	0.27
*S. albicans*	7/62	11.3%	7/68	10.3%	14/130	10.8%	0.992
Other	13/62	21.0%	13/68	19.1%	26/130	20.0%	0.989
Mixed Growth	8/178	4.5%	1/179	0.6%	9/357	2.5%	0.02
Fungi	7/178	3.9%	2/179	1.1%	9/357	2.5%	0.104
Yeast	5/7	71.4%	1/2	50.0%	6/9	66.7%	0.101
Filamentous	2/7	28.6%	1/2	50.0%	3/9	33.3%	0.56
Acanthamoeba	2/178	1.1%	5/179	2.8%	7/357	2.0%	0.448
Viruses	4/178	2.2%	6/179	3.4%	10/357	2.8%	0.75
HSV	3/4	75.0%	1/6	16.7%	4/10	40.0%	0.315
VZV	1/4	25.0%	3/6	50.0%	4/10	40.0%	0.32
*Adenovirus*	0/4	0.0%	2/6	33.3%	2/10	20.0%	0.158

## Data Availability

The original contributions presented in this study are included in the article. Further inquiries can be directed to the corresponding author.

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
