# Peer review of "Five-Year Analysis of Microbial Keratitis Incidence, Isolates, and In Vitro Antimicrobial Sensitivity in the South West of England: An Epidemiological Study"

_microorganisms, 2025, doi:10.3390/microorganisms13071578_

Round 1

Reviewer 1 Report

Comments and Suggestions for Authors

General comments:

This is s a retrospective study on microbial keratitis in Southwest of England. The presented data have serious limitations that partially have been discussed and revealed in the manuscript. This is therefore rather epidemiological analysis than scientific work and should be classified properly. There are some updates that should be implemented to increase the value of the manuscript. Details are listed in the specific sections below. 

First of all, please use correct names of the species/genera. I.e. there is no “Staph Epidermidis” in the nomenclature. It should be S. epidermidis or full name Staphylococcus epidermidis and so on. Please use italics correctly, i.e. “in vitro” should be in vitro, ..these are really basics. 

Material and methods:

Please provide rationale/background for having these data split into 2 groups. Why not 1 group or 3 groups? Why these periods? Is this somehow connected with the COVID-19 pandemia?

You stated: “Organisms were classified as gram-positive bacteria, gram-negative bacteria, acanthamoeba, viruses or fungi.” However there is lack of methodology for the species/genera classification. Please update.

Results:

Table 1. – needs revision per organism’s names (italics, correct nomenclature). Please also adjust the table so data are clearly assigned to the years (paste inside vertical borders?)

Figure 2 - needs revision per organism’s names (italics, correct nomenclature).

Please define/precise “susceptibility” and “sensitivity” terms used in this manuscript.

Discussion:

Corneal scrape outcomes and Limitations part : I would go deeper into the fact that in >60% of scrapings the etiological agent was not classified or was unknown. This is the weakest point of the presented data as we may assume that organisms that are recognized are the easiest to culture/classification per methodology used at these hospitals. So, the methodology used narrows the outcome we get.  This applies especially to viruses, fungi and protozoans which are less common and less diagnosed due to this – vicious circle. Please discuss a bit more about this phenomenon.

Incidence of microbial keratitis part: You suggested that incidence of MK correlates with the contact lenses use, but you have not shown any correlation in data presented. There is no information on the percentage of contact lenses users as well. May you please update this or provide some explanation or citations that would support this possible explanation?

Comments on the Quality of English Language

Quality of the nomenclature used and results presentation must be improved.

Author Response

Comment 1: This is s a retrospective study on microbial keratitis in Southwest of England. The presented data have serious limitations that partially have been discussed and revealed in the manuscript. This is therefore rather epidemiological analysis than scientific work and should be classified properly. There are some updates that should be implemented to increase the value of the manuscript. Details are listed in the specific sections below. 

We agree therefore we have reclassified this as an epidemiological study. Please see the title and line 70 in materials and methods.

Comment 2: First of all, please use correct names of the species/genera. I.e. there is no “Staph Epidermidis” in the nomenclature. It should be S. epidermidis or full name Staphylococcus epidermidis and so on. Please use italics correctly, i.e. “in vitro” should be in vitro, ..these are really basics. 

Agree. Organism names have been corrected and typed in the correct font throughout the manuscript. Similarly all uses of the word 'Gram' as in Gram stain and gram-positive etc have been corrected.

Comment 3: Please provide rationale/background for having these data split into 2 groups. Why not 1 group or 3 groups? Why these periods? Is this somehow connected with the COVID-19 pandemia?

Please see Statistical analysis, line 106

Comment 4: You stated: “Organisms were classified as gram-positive bacteria, gram-negative bacteria, acanthamoeba, viruses or fungi.” However there is lack of methodology for the species/genera classification. Please update.

I have tried to explain the methodology more clearly in lines 94-96. I was unsure of what exactly the reviewer was asking for in this comment. 

Comment 5: Table 1. – needs revision per organism’s names (italics, correct nomenclature). Please also adjust the table so data are clearly assigned to the years (paste inside vertical borders?)

Done.

Comment 6: Figure 2 - needs revision per organism’s names (italics, correct nomenclature).

Done.

Comment 7: Please define/precise “susceptibility” and “sensitivity” terms used in this manuscript.

I have used the term sensitivity throughout and omitted the term susceptibility in order to avoid confusion. Both words are interchangeable. I have added a definition of 'sensitivity' in line 59.

Comment 8: Corneal scrape outcomes and Limitations part : I would go deeper into the fact that in >60% of scrapings the etiological agent was not classified or was unknown. This is the weakest point of the presented data as we may assume that organisms that are recognized are the easiest to culture/classification per methodology used at these hospitals. So, the methodology used narrows the outcome we get.  This applies especially to viruses, fungi and protozoans which are less common and less diagnosed due to this – vicious circle. Please discuss a bit more about this phenomenon.

Agree. I have added another paragraph addressing this in line 191 of the discussion and line 308 of the limitations section.

Comment 9: Incidence of microbial keratitis part: You suggested that incidence of MK correlates with the contact lenses use, but you have not shown any correlation in data presented. There is no information on the percentage of contact lenses users as well. May you please update this or provide some explanation or citations that would support this possible explanation?

Agree. I have added relevant references to support this in line 211. Unfortunately data collection did not include history of contact lens wear. I was not responsible for data collection.

Reviewer 2 Report

Comments and Suggestions for Authors suggestions for the author:

1. in line 54 and 75 specify what type of microscope is used,

2. in line 120 enter the percentage corresponding to the number of cases

fouthy four

3. in line 196 enter the number of culture positive

question: 

out of 873 scrapes, only 357 cultures are positive, the rest are negative or no cultures have been carried out or anything else

Author Response

Comment 1: 1. in line 54 and 75 specify what type of microscope is used,

I do not have the answer to this as I have never visited the microbiology lab.

Comment 2: 2. in line 120 enter the percentage corresponding to the number of cases

fouthy four

Corresponding number of cases added (line 115)

Comment 3: 3. in line 196 enter the number of culture positive

Done. Line 241.

Comment 4: question: 

out of 873 scrapes, only 357 cultures are positive, the rest are negative or no cultures have been carried out or anything else

Explanation added in line 123

Reviewer 3 Report

Comments and Suggestions for Authors

I consider that this is a review of specific cases that shows a general overview of ocular pathology in a specific region, which is well described and can support the addition of epidemiological data in future reports from other regions and countries.

Author Response

N/A

Reviewer 4 Report

Comments and Suggestions for Authors

The manuscript presents a well-structured retrospective study on microbial keratitis in Southwest England. The data provided are valuable and contribute to the understanding of regional antimicrobial resistance trends.

The increase in Staphylococcus epidermidis should be discussed with potential explanations.

Author Response

The increase in Staphylococcus epidermidis should be discussed with potential explanations

I have commented that the number of cultures positive for s. epidermidis is extremely low in group A and that it is difficult to draw any significant conclusion here given that even a small increase would be significant. 

Reviewer 5 Report

Comments and Suggestions for Authors

The scientific literature contains numerous studies that analyze various aspects of microbial keratitis (MK) worldwide: frequency, risk factors, pathogenesis, diagnostic tools, treatment, costs, management, MK analysis in the COVID-19 pandemic, contact lens-associated MK, etc...

Many retrospective studies (6, 10, or 12-year periods) have analyzed MK patterns in the UK, globally, or in selected regions (North, East, North-eastern regions, Scotland, Nottingham, Oxford, London). Data were collected from various tertiary hospitals, royal hospitals, etc.

The present study is a five-year analysis (2018-2022) of microbial keratitis incidence, isolates, and in-vitro antimicrobial sensitivity in the Southwest of England. The authors collected data from 4 hospitals, and their study appears to be helpful only for local healthcare practitioners and ophthalmologists, as shown in lines 61-63. 

The reviewer carefully examined the current version, and the following comments are available below:

  1. Abstract

It is cumbersome and blurred in the current version. 

The authors are invited to organize the abstract into background/objective, methods, results, and conclusions. They should also select significant keywords linked to the particular data presented. 

2. Introduction

No new and specific data are provided in the current version.

The authors are invited to perform a literature review and present the findings of previous studies about MK in the  UK. Thus, they can attract the readers' interest with an extensive overview of specific aspects of MK in the UK from different points of view: epidemiology, incidence (correlated with sex, residence, age, associated diseases, hygiene, contact lens, etc.) risk factors, the most involved pathogens and their AMR, treatment protocol, social and professional burden, etc.

Afterward, they can present the hypotheses that underline the present study and its novelty and potential applications.

3. Materials and methods

To attract more readers' interest, the authors are invited to present more sociodemographic data about the patients included in their retrospective study: age, residence, comorbidities, contact lens wearing (Yes/No), education, and occupation. Thus, they can correlate the MK incidence with all these baseline data.

4. Results

The authors are invited to correctly mention the pathogens' scientific names in both tables and in the MS text. What is CoNS mentioned in Tables 1 and 2? 

They should also present the original results in detail and show the significant differences.

The authors are invited to show the therapeutic protocol applied to MK patients in the mentioned hospitals and the disease evolution, complications, and other outcomes.

Then, they are invited to correlate the laboratory results with the suggested baseline data and provide a complex overview of MK epidemiology. 

5. Discussion

The authors are invited to explain the reason for the present study design and motivate the period separation (2018-2020 and 2021-2022).

The original results are not discussed and explained enough in the current MS version. 

Moreover, the authors should compare their results with those reported in various UK regions. 

Numerous other limitations linked to the lack of sociodemographic and baseline data must be mentioned.  

6. After all, the essential findings, applications, and further directions should be presented in the Conclusions.

As an overview, the current form needs extensive and rigorous revision, according to the above-mentioned comments. The same recommendation is available for English and MS data editing and presentation.

Comments on the Quality of English Language

major revision

Author Response

  1. Abstract

It is cumbersome and blurred in the current version. 

The authors are invited to organize the abstract into background/objective, methods, results, and conclusions. They should also select significant keywords linked to the particular data presented. 

Answer: the abstract has been organised properly and keywords altered to reflect the paper more accurately

2. Introduction

No new and specific data are provided in the current version.

The authors are invited to perform a literature review and present the findings of previous studies about MK in the  UK. Thus, they can attract the readers' interest with an extensive overview of specific aspects of MK in the UK from different points of view: epidemiology, incidence (correlated with sex, residence, age, associated diseases, hygiene, contact lens, etc.) risk factors, the most involved pathogens and their AMR, treatment protocol, social and professional burden, etc.

Afterward, they can present the hypotheses that underline the present study and its novelty and potential applications.

Answer: a literature review would go beyond the scope of this article. We wished to solely focus on this region in the UK but we have made comparisons to other studies.

3. Materials and methods

To attract more readers' interest, the authors are invited to present more sociodemographic data about the patients included in their retrospective study: age, residence, comorbidities, contact lens wearing (Yes/No), education, and occupation. Thus, they can correlate the MK incidence with all these baseline data.

Answer: unfortunately I was not involved in the data collection. Having gone through the data that was passed on to me, there is no useful demographic data. Only one co-author commented on contact lens wear at their hospital.

4. Results

The authors are invited to correctly mention the pathogens' scientific names in both tables and in the MS text. What is CoNS mentioned in Tables 1 and 2? 

Answer: Pathogen names have been entered correctly throughout the manuscript. I have added a key above each table to explain what CoNS stands for.

They should also present the original results in detail and show the significant differences.

Answer: I have added in an additional table (table 4) with more of the original results specific to important organisms and antibiotic classes. The raw data is cumbersome and lengthy and would not look aesthetically pleasing in the manuscript. 

The authors are invited to show the therapeutic protocol applied to MK patients in the mentioned hospitals and the disease evolution, complications, and other outcomes.

Answer: added in line 98 of the methods section. I have no data on disease evolution/complications or other outcomes as the data was collected by other authors.

Then, they are invited to correlate the laboratory results with the suggested baseline data and provide a complex overview of MK epidemiology. 

Answer: again, I do not have access to this data as I did not collect any data myself.

5. Discussion

The authors are invited to explain the reason for the present study design and motivate the period separation (2018-2020 and 2021-2022).

Answer: explained in statistical analysis line 105

The original results are not discussed and explained enough in the current MS version. 

Answer: I have delved deeper into discussion of the results in the discussion section. Please see additional paragraphs at lines 193, 213, 251, 278.

Moreover, the authors should compare their results with those reported in various UK regions. 

Numerous other limitations linked to the lack of sociodemographic and baseline data must be mentioned. 

Answer: I have responded to this in the limitation section (line 306)

6. After all, the essential findings, applications, and further directions should be presented in the Conclusions.

Answer: the conclusion has been altered appropriately with suggestions for further research and changes within microbiology labs.

Round 2

Reviewer 5 Report

Comments and Suggestions for Authors

The reviewer appreciates the author's efforts to revise the MS according to the previous review report. However, not all requests received suitable responses. 

The following comments are available below:

Abstract:

Line 31: Pathogen is better than organism - please check and correct. The same for Tables 1, 2, and 4 caption and the first column header.

Line 40: Keywords: microbial keratitis without " ; " between the 2 words; please verify and correct.

Introduction

No changes were made in this section; the authors responded as follows: 

"A literature review would go beyond the scope of this article. We wished to solely focus on this region in the UK, but we have made comparisons to other studies."

However, most recent articles published in highly ranked academic journals deeply show previous findings in the introduction to attract readers' interest, demonstrate the novelty of their work, and the capacity to enrich the scientific database with new and essential information. 

Materials and Methods

The requested data is not available. The lack of data is explained as follows: "I have no data on disease evolution/complications or other outcomes as the data was collected by other authors." Are they not included in the current team to ensure the requested data aimed at improving MS quality?

The current form contains mixed data. Please organize them into subsections and detail each item.

Line 86-90. Please provide information from all medical centers and put them in a subsection.

Statistical analysis: Please indicate the software and tools used for Statistics.

Results and Discussions

Table 1 - please verify and use italics for all pathogen species: Streptococcus, Bacillus

Mixed growth is included in total pathogens?

Please revise all table formats, verify the headers, standardize the characters, and make them in the same format as the MDPI instructions request. 

Please maintain the same trend in the results' presentation, before/after the pandemic; the same comment is available for Discussions.

Please present the limitations more concisely.

Comments on the Quality of English Language

Moderate revision.

Author Response

Comment 1:

Line 31: Pathogen is better than organism - please check and correct. The same for Tables 1, 2, and 4 caption and the first column header. 

This has been corrected

Comment 2:

Line 40: Keywords: microbial keratitis without " ; " between the 2 words; please verify and correct.

This has been corrected.

Comment 3:

Introduction

No changes were made in this section; the authors responded as follows: 

"A literature review would go beyond the scope of this article. We wished to solely focus on this region in the UK, but we have made comparisons to other studies."

However, most recent articles published in highly ranked academic journals deeply show previous findings in the introduction to attract readers' interest, demonstrate the novelty of their work, and the capacity to enrich the scientific database with new and essential information. 

I have now added sections on aetiology and epidemiology of MK and antimicrobial susceptibility in the introduction. I have referenced previous large scale studies in order to make this paper more useful and engaging for readers.

Comment 4:

The requested data is not available. The lack of data is explained as follows: "I have no data on disease evolution/complications or other outcomes as the data was collected by other authors." Are they not included in the current team to ensure the requested data aimed at improving MS quality?

I have been in communication with the other authors but only one author kept a record on contact lens wear in subjects. Barring age and gender there was no other demographic data that was collected on patients. 

Comment 5:

The current form contains mixed data. Please organize them into subsections and detail each item.

I have now organised the materials and methods into a more structured format with subheadings to make it easier for the reader to understand.

Comment 6:

Line 86-90. Please provide information from all medical centers and put them in a subsection.

I have created a subsection outlining hospital scraping protocols (line 154). I have also added a line on hospital treatment protocols (line 254)

Comment 7:

Statistical analysis: Please indicate the software and tools used for Statistics.

Line 260 - I have made this clearer now.

Comment 8:

Results and Discussions

Table 1 - please verify and use italics for all pathogen species: StreptococcusBacillus - done

Mixed growth is included in total pathogens? - yes it is, mixed growth was only counted as one isolate. In line 288 I have mentioned that mixed growth was bacterial only (i.e. there were no isolates consisting of bacteria and another type of pathogen such as an amoeba or fungus)

Please revise all table formats, verify the headers, standardize the characters, and make them in the same format as the MDPI instructions request. - done

Please maintain the same trend in the results' presentation, before/after the pandemic; the same comment is available for Discussions. - I have now made more comparisons between the data for groups A and B and commented on trends between the two groups.

Please present the limitations more concisely. - done

I have submitted the new manuscript with changes tracked to help the reviewers see which changes have been made.